# Tumor Necrosis Factor-Related Apoptosis-Inducing Ligand (TRAIL) in Patients after Acute Stroke: Relation to Stroke Severity, Myocardial Injury, and Impact on Prognosis

**DOI:** 10.3390/jcm11092552

**Published:** 2022-05-02

**Authors:** Michal Mihalovic, Petr Mikulenka, Hana Línková, Marek Neuberg, Ivana Štětkářová, Tomáš Peisker, David Lauer, Petr Tousek

**Affiliations:** 1Cardiocenter, Third Faculty of Medicine, Charles University, 100 34 Prague, Czech Republic; 2Department of Neurology, Third Faculty of Medicine, University Hospital Kralovske Vinohrady, Charles University, 100 34 Prague, Czech Republic; 3Medtronic Czechia, Partner of INTERCARDIS Project, 190 00 Prague, Czech Republic

**Keywords:** acute stroke, TRAIL, mortality, cardiovascular disease

## Abstract

Background: Tumor necrosis factor-related apoptosis-inducing ligand (TRAIL) is known to be associated with poor prognosis after cardiovascular events. We aimed to assess the dynamic changes in TRAIL levels and the relation of TRAIL level to stroke severity, its impact on the short-term outcomes, and its association with markers of cardiac injury in patients after acute stroke. Methods: Between August 2020 and August 2021, 120 consecutive patients, 104 after acute ischemic stroke (AIS), 76 receiving reperfusion therapy, and 16 patients after intracerebral hemorrhage (ICH) were enrolled in our study. Blood samples were obtained from patients at the time of admission, 24 h later, and 48 h later to determine the plasma level of tumor necrosis factor-related apoptosis-inducing ligand (TRAIL), N-terminal prohormone of brain natriuretic peptide (NT-proBNP), and high-sensitive Troponin I (hs-TnI). Twelve-lead ECGs were obtained at the time of admission, 24 h later, 48 h later, and at the release of the patients. Evaluations were performed using the National Institutes of Health Stroke Scale (NIHSS) at the time of admission and using the modified Rankin Scale (mRS) 90 days following the patient’s discharge from the hospital. Results: We observed a connection between lower TRAIL levels and stroke severity evaluated using the NIHSS (*p* = 0.044) on the first day. Lower TRAIL showed an association with severe disability and death as evaluated using the mRS at 90 days, both after 24 (*p* = 0.0022) and 48 h (*p* = 0.044) of hospitalization. Moreover, we observed an association between lower TRAIL and NT-proBNP elevation at the time of admission (*p* = 0.039), after 24 (*p* = 0.043), and after 48 h (*p* = 0.023) of hospitalization. In the ECG analysis, lower TRAIL levels were associated with the occurrence of premature ventricular extrasystoles (*p* = 0.043), and there was an association with prolonged QTc interval (*p* = 0.052). Conclusions: The results show that lower TRAIL is associated with stroke severity, unfavorable functional outcome, and short-term mortality in patients after acute ischemic stroke. Moreover, we described the association with markers of cardiac injury and ECG changes.

## 1. Introduction

Tumor necrosis factor-related apoptosis-inducing ligand (TRAIL) is a member of the tumor necrosis factor ligand superfamily which is expressed as a transmembrane protein on the cell surface of a variety of cell types or is released as a soluble protein. It is described as a protein that mediates cell signals through its receptors, TRAIL-R1/DR4 and TRAIL-R2/DR5, triggering caspase activation and programmed cell death in several cell types. However, TRAIL also has the ability to bind two decoy receptors, TRAIL-R3/DcR1 and R4/DcR2 and Osteoprotegerin, therefore revealing a more complex system with a wide range of biological effects [1]. Its biological functions have predominantly been focused on its anti-cancer activity [2,3]. However, in recent years, growing attention is focused on its involvement in several pathological conditions and its different effects, confirming the ambiguity of this protein. TRAIL, with its receptors, is expressed in various cells including endothelial cells, vascular smooth muscle cells, and inflammatory cells [4,5]. Kakarenko et al. described the involvement of TRAIL in cardiovascular diseases [6]. In patient studies, a lower plasma TRAIL level was associated with acute myocardial infarction (AMI) and with worse left ventricular ejection fraction in ST-elevation AMI [7]. TRAIL was described as a potential predictor of poor prognosis in patients after AMI, with coronary artery disease (CAD) or advanced heart failure [8,9,10,11,12,13]. Moreover, a lower TRAIL level was described as a prognostic biomarker for atheromatosis plaque formation in patients with chronic kidney disease [14]. Volpato et al. demonstrated that the risk of death increased in patients with prevalent cardiovascular disease with a lower TRAIL level compared to those with higher values. Furthermore, all-cause and cardiovascular mortality increased, with a very high mortality risk associated with the lowest TRAIL levels [8,13]. The involvement of TRAIL in the pathophysiology of cardiovascular diseases as a possible executor of neuronal and myocardial apoptosis was reported [15,16,17]. It was described that TRAIL levels in acute stroke patients were significantly lower compared to the standard population [18]. Moreover, TRAIL levels were significantly lower in large artery atherosclerosis stroke patients, and this was associated with increased severity of cerebral atherosclerosis [19]. Moreover, Kang et al. showed an association between lower TRAIL levels, NIHSS score, and stroke volume [20]. Stroke is a major public health problem, and it is associated with socioeconomic burden because patients often suffer functional impairment and disability. The acute phase after stroke is critical with the necessity of rapid response and fast management [21]. In this light, this study aimed to describe the dynamic changes in TRAIL levels in patients in the acute phase after stroke and its association with stroke severity, impact on the short-term outcomes, and prevalence of cardiovascular involvement.

## 2. Materials and Methods

### 2.1. Study Design and Patients

The study population consisted of consecutive patients with acute ischemic stroke (AIS) or intracerebral hemorrhage (ICH) who were enrolled in the Department of Neurology, University Hospital Kralovske Vinohrady, Prague. Between August 2020 and August 2021, 120 consecutive patients after AIS and ICH were enrolled in our study. Inclusion criteria were AIS or ICH diagnosed on clinical and non-contrast head CT results, supplemented by CT angiography or magnetic resonance. All available clinical data and other predictor variables (demographics, hemodynamics, and blood results) were obtained. Standard neurologic examination was performed, stroke severity was assessed using the National Institutes of Health Stroke Scale (NIHSS) at the time of admission, defined with the scale: no stroke symptoms (0), minor (1–4), moderate (5–15), moderate to severe (16–20), and severe (21–42). Functional outcome was evaluated using the modified Rankin Scale (mRS) including death within the first 90 days, defined as the categories: no symptoms (0), no disability despite symptoms (1), slight disability (2), moderate (3), moderate to severe (4), severe (5), and death (6). The study was approved by the local Ethics Committee, and written informed content was obtained from each patient. The study protocol conforms to the ethical guidelines of the 1975 Declaration of Helsinki. Exclusion criteria were the following: acute cardiac decompensation, significant valve disease, and acute myocardial infarction.

### 2.2. Laboratory Analysis

Venous blood samples were obtained from patients after acute stroke within the first 48 h: at the time of admission (day 0) and 24 ± 12 (day 1) and 48 ± 12 (day 2) hours later. Collected blood was separated by centrifugation of the blood (3500 rpm, 15 min) and afterwards stored at −70 °C before analysis. Serum TRAIL levels were measured using a commercially available enzyme-linked immunosorbent assay (ELISA) kit (R&D Systems, Minneapolis, MN), following the manufacturer’s instructions, and analyzed with an ELISA reader at 450 nm. All samples were measured in duplicate and averaged. The sensitivity of the assay was 2.86 pg/mL, and the intra- and inter-assay coefficients of variation (CV) were 5.6% and 7.4%, respectively, and the upper limit of detection was 1000 pg/mL. The cut-off value for lower plasma TRAIL levels was set as 64 pg/mL; the level was defined from the median TRAIL level in our patients and published studies as compared with the control group [8,20]. Blood samples for hematology (e.g., hemoglobin level, leukocyte count, and platelet count) and biochemistry analysis of high-sensitive troponin I (hs-cTnI) (cut off in our hospital 53 ng/l), N-terminal prohormone of brain natriuretic peptide (NT-proBNP), potassium, and CRP levels were obtained at the same time as TRAIL. 

### 2.3. Electrocardiogram (ECG) and Echocardiographic Analysis

Twelve-lead ECGs were obtained at the time of admission, 24 ± 12 (day 1) hours later, 48 ± 12 (day 2) hours later, and at the discharge of the patients by the nurses if the patients were eligible. The ECGs were analyzed by one observer who was blinded to all clinical data. The following changes were recorded when abnormal: atrial fibrillation, atrial flutter, sinus tachycardia: HR > 120, sinus bradycardia: HR < 45, atrioventricular block, first-, second-, and third-degree, ventricular tachycardia (more than three beats of ventricular origin), ectopic ventricular beats, ST-elevation, ST-depression, isoform T-wave, inverted T-wave, U-wave, and QTc > 0.45 s for men and >0.46 s for women. An echocardiographic examination was performed within the first 5 days of hospitalization if patients were eligible.

### 2.4. Statistical Analysis

Categorical variables were recorded as frequencies or counts (percentages). Continuous data were tested for distribution using the Kolmogorov–Smirnov test. Continuous variables are presented in graphs and tables as mean and standard deviation. Parameters that did not follow a normal distribution were analyzed with the Mann–Whitney U test or the Kruskal–Wallis test and expressed as the median. For continuous variables, parameters that followed normal distribution were analyzed with Student’s t test and described as the mean ± standard deviation. A chi-square test or Fisher’s exact test was used to detect the difference between categorical variables. Results were considered statistically significant at a significance level of *p* < 0.05. All statistical analyses were performed in IBM SPSS Statistics version 26. Graphical analyzes were performed in Sigma plot version 14.

## 3. Results

### 3.1. Patients’ Population

The total patient study population consisted of 120 patients (63 men and 57 women) with a mean age of 70.9 ± 13.2 years, and 104 with AIS and 16 with ICH were included in the study. In the AIS group, 73% of patients received reperfusion therapy (29 patients underwent mechanical trombectomy and 69 patients received intravenous trombolysis). The baseline characteristics are summarized in Table 1. The most common comorbidities included arterial hypertension, dyslipidemia, DM 2 type, and atrial fibrillation. We did not observe severe hyperglycaemia or septic condition in the first 48 h. Except for one, all patients were admitted within 12 h of the onset of symptoms. In the AIS group, 78.8% of patients presented with minor to moderate stroke (NIHSS 1–15), and 21.2% presented with moderate to severe or severe stroke (NIHSS 16–42) at the time of admission. In the AIS group, 89 patients (85.6%) underwent an echocardiographic examination within 5 days of hospitalization, and 2 patients presented with new left ventricular regional wall motion abnormality. Nine patients (8.7%) in the AIS group and eight patients (43%) in the ICH group died during hospitalization. 

### 3.2. Characteristics of TRAIL and Its Dynamic Changes after Acute Stroke

Values of TRAIL at different time points during the first 48 h in patients with AIS and ICH are shown in Figure 1. The mean TRAIL level during the first 48 h of hospitalization was 72.58 ± 32.81 pg/mL in the AIS group and 55.83 ± 30.66 pg/mL in the ICH group. The mean TRAIL level decreased in the first 24 h of hospitalization and subsequently increased on day 2. We observed the lowest TRAIL level in both groups on day 1 with the mean level of 68.73 ± 33.42 in the AIS group and 50.12 ± 27.35 pg/mL in the ICH group. The TRAIL level was lower in the ICH group; we observed differences between the TRAIL levels in the AIS group and the ICH group on day 1 (*p* = 0.03) and day 2 (*p* = 0.034) (Figure 1). Lower TRAIL levels were associated with the presence of DM 2 type (*p* = 0.046) and smoking (*p* = 0.026).

### 3.3. TRAIL and NT-proBNP Are Associated with Stroke Severity and Worse Neurological Outcome in AIS

Patients with moderate to severe and severe stroke (NIHSS 16–42) presented with lower TRAIL levels than patients with minor to moderate stroke (73.1 pg/mL vs. 51.3 pg/mL, *p* = 0.003) (Figure 2a). Regarding the mRS 90-day score of functional disability, patients with worse functional outcome or death (mRS 90 5–6) presented with lower TRAIL levels (72.6 pg/mL vs. 43.1 pg/mL, *p* < 0.001) (Figure 2b). 

We observed a connection between lower TRAIL levels and moderate to severe and severe stroke (NIHSS 16–42) on day 1 (*p* = 0.044) (Figure 3a). Moreover, lower TRAIL was associated with severe disability or death on both day 1 (*p* < 0.0022) and day 2 (*p* < 0.044) (Figure 3b). A lower TRAIL level on day 1 was associated with mortality at 90 day (*p* = 0.009). In the ICH group, we did not observe an association between lower TRAIL levels and worse functional outcome on day 0 (*p* = 0.58), day 1 (*p* = 0.24), or day 2 (*p* = 0.59); however, the patients’ frequency was very low.

Furthermore, we observed an association between elevated NT-proBNP > 125 pg/mL and moderate to severe or severe stroke both at the time of admission (*p* = 0.046) and on day 1 (*p* = 0.002). Elevated NT-proBNP was associated with worse functional disability evaluated using the mRS 90-day score at the time of admission (*p* = 0.0014), on day 1 (*p* = 0.0002), and day 2 (*p* = 0.034). We did not find an association between lower TRAIL levels and the time of admission to the hospital later than 4.5 h (12.5%, *p* = 0.49), 12 h (1%, *p* = 0.48), or with wake-up stroke (7.4%, *p* = 0.58).

### 3.4. TRAIL and Markers of Myocardial Injury in Patients after Acute Stroke

Several studies have reported possible cardiac injury in patients after acute stroke. Myocardial injury in patients after acute stroke is believed to be caused either by ischemic injury in patients with concomitant coronary artery disease or direct injury through an autonomic imbalance known as neurogenic heart syndrome [22]. Cardiac troponins are sensitive biomarkers widely used for the diagnosis of acute myocardial infarction. Several studies reported troponin elevation in patients after stroke and its prognostic information for short- and long-term outcomes and survival [23,24]. NT-proBNP is the N-terminal fragment of B-type natriuretic peptide secreted by myocytes as a reaction to wall stress. It was described as a predictor in diagnosis and prognosis in patients with heart failure, left ventricular dysfunction, and coronary syndromes [25,26,27]. Several studies showed that NT-proBNP is also associated with the risk of ischemic and hemorrhagic stroke [28,29].

In our group, 44.2% of the patients in the AIS group and 50% in the ICH group presented with elevated NT-proBNP >125 pg/mL during the first 48 h of hospitalization. In the AIS group, the highest average value was reached on day 1 of hospitalization.

Elevated high-sensitive troponin I (hs-cTnI) was presented in 22.1% of the patients in the AIS group and 12.5% in the ICH group during the first 48 h of hospitalization. In the AIS group, lower TRAIL levels were connected with elevated NTproBNP at the time of admission (*p* = 0.039), after 24 h (*p* = 0.043) and after 48 h (*p* = 0.023) (Figure 3c). We found no association between lower TRAIL levels and elevated hs-cTnI (Figure 3d). The development of cardiac markers and TRAIL level changes in the ICH group are shown in Figure 4a,b.

### 3.5. TRAIL and ECG Changes in AIS

In the ECG analysis, the most common ECG changes included QTc prolongation (22.1%), ST depression (18.2%), and T wave inversion (15.4%). Association between lower TRAIL level and ECG changes is summarized in Table 2. We found no association between lower levels of TRAIL and morphological changes. We observed near to significant association with prolonged QTc interval (*p* = 0.052). Atrial fibrillation was the most common arrhythmia, occurring in almost 1/3 of the patients during the first 48 h of hospitalization. However, there was no association between lower levels of TRAIL and the occurrence of atrial fibrillation. We found an association between lower TRAIL levels and the occurrence of premature ventricular contractions (*p* = 0.043).

## 4. Discussion

In this study, we showed that low TRAIL levels are associated with stroke severity, unfavorable short-term outcome, and markers of cardiac injury in patients after acute stroke. TRAIL levels seem to represent an important predictor of prognosis, with low TRAIL levels suggesting poor prognosis. Several studies have shown an association between lower TRAIL levels and a higher risk of death [8,11,30]. Kang et al. showed an association between lower TRAIL levels and NIHSS score, with an increased relative risk for patients with serum TRAIL < 64.0 pg/mL for the presence of an NIHSS score 6–24 [20]. Volpato et al. demonstrated that the risk of death increased 3-fold in patients with prevalent cardiovascular disease with the lowest quartile TRAIL levels (<59 pg/mL) compared to the highest values. Furthermore, all-cause and cardiovascular mortality increased with TRAIL levels near 70 pg/mL, with very high mortality risk associated with the lowest TRAIL levels [8]. As our study suggests, TRAIL levels decrease in the acute phase, reaching their lowest levels within the first 24 h of hospitalization with a subsequent increase. This shows a similar pattern as seen in previous studies, where TRAIL is decreased shortly after acute myocardial infarction and increased in the following days [10,11]. 

There are several possible mechanisms by which TRAIL is involved in pathophysiology in patients after stroke. Acute stroke leads to a cascade of pathological reactions leading to pathophysiological alteration, neuronal cell death, and local and systemic inflammation. Neuronal damage and molecules released from damaged cells lead to local inflammation and the activation of the microglia. This is followed by leukocyte recruitment to the lesion contributing to tissue injury, where TRAIL contributes to the pathogenesis of apoptosis and the inflammation processes [31,32,33].

Furthermore, to our knowledge, this is the first report to evaluate the association of TRAIL levels and markers of cardiac injury in patients after acute stroke. Studies have shown that patients after acute stroke are often presented with signs of cardiac injury including troponin and NT-proBNP elevation, ECG changes, and in some cases echocardiographic changes. Cardiomyocyte injury in patients after acute stroke was previously described to be possibly caused by autonomic dysfunction and higher catecholamine levels, which may lead to a decrease in TRAIL levels through the beta-adrenergic receptor [34,35,36,37]. This is supported by a study showing that impairment of the beta-adrenergic receptor and its upregulation may be responsible for synergic cell death observed with TRAIL [38]. Other potential mechanisms include cleavage by metalloproteinase 2 which is upregulated, for example, in acute myocardial infarction (AMI) and can cleave TRAIL [39]. This suggests that TRAIL levels could be influenced by a combination of pathological reactions involving the autonomous nervous system, neuronal apoptosis, and systemic and local inflammation response. Moreover, studies have shown a protective effect of TRAIL against apoptosis and an effect on the proliferation of endothelial cells by activating Akt and ERK pathways [40]. Zauli et al. showed that TRAIL upregulates nitric oxide and prostanoid production in primary human endothelial cells which play important roles in endothelial cell function [41]. Furthermore, protective effects against atherosclerosis progression and anti-inflammatory effects have been described [41,42,43]. 

Dynamic changes and TRAIL decreases in the acute phase of pathological conditions such as acute stroke or AMI are not fully understood. However, studies suggest that TRAIL reduction might represent an acute consequence of consumption during anti-inflammatory effects, protective endothelial effects, apoptosis, autonomic dysregulation, and cardiovascular alteration with subsequent recovery, suggesting that low levels of TRAIL could reduce inhibition power against apoptosis [8]. Therefore, TRAIL could also hide therapeutic possibilities that should be addressed in further studies.

There are some limitations to the performed study. Our study was only focused on the acute phase after stroke, and the study group was heterogenous regarding the type of reperfusion therapy. Acute infection and severe hyperglycemia were not part of the exclusion criteria. Moreover, the ECG monitoring length depended on the clinical condition of the patients, which could lead to lower PVC prevalence. Lastly, setting a lower TRAIL level cut-off is a challenge. Currently, there is no consensus on the range of TRAIL in patient studies, which may make interpretation difficult. In our study, we based the cut-off for TRAIL levels on comparisons with other studies and the median TRAIL level of our patients.

In conclusion, the functions of TRAIL are complex. It is involved in multiple pathways depending on the specific cell type and the specific pathological situations. A lower TRAIL level seems to represent a negative prognostic marker for all-cause mortality and worse clinical outcomes after pathological processes. We reported that low TRAIL levels represent a predictor of greater stroke severity and poorer short-term outcomes in patients after acute stroke. Moreover, it provides additional information regarding the relationship between markers of cardiac injury in patients after acute stroke as a potential outcome of cardiovascular disease. Therefore, the elucidation of the concept and ambiguity of this protein may help us in understanding the development of several diseases and could lead to new therapeutic approaches.

## Figures and Tables

**Figure 1 jcm-11-02552-f001:**
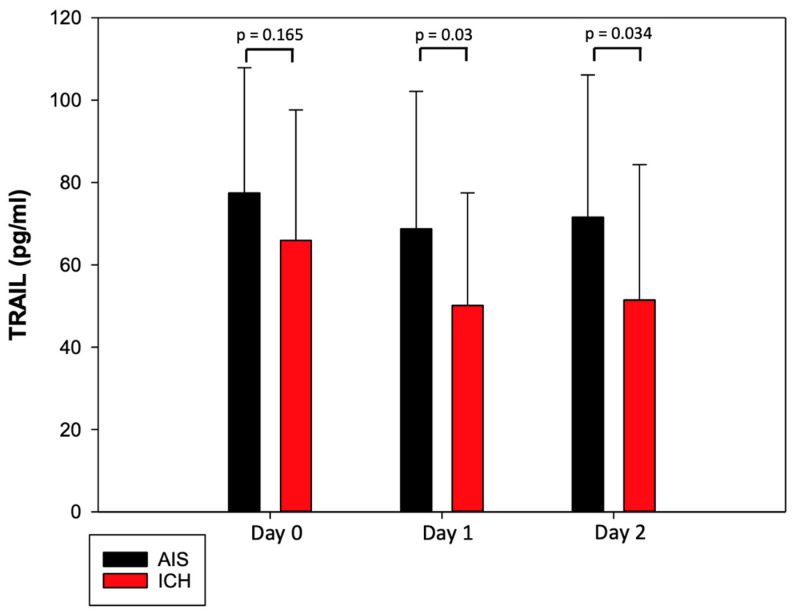
TRAIL levels during first 48 h of hospitalization in patients after acute ischemic stroke and intracerebral hemorrhage.

**Figure 2 jcm-11-02552-f002:**
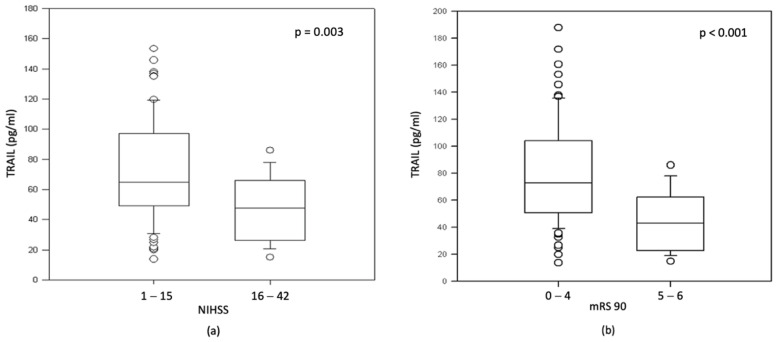
Association between TRAIL levels on day 1 and NIHSS score and mRS at 90 days in patients after AIS. (**a**) Association between TRAIL level and NIHSS score; (**b**) association between TRAIL level and mRS at 90 days.

**Figure 3 jcm-11-02552-f003:**
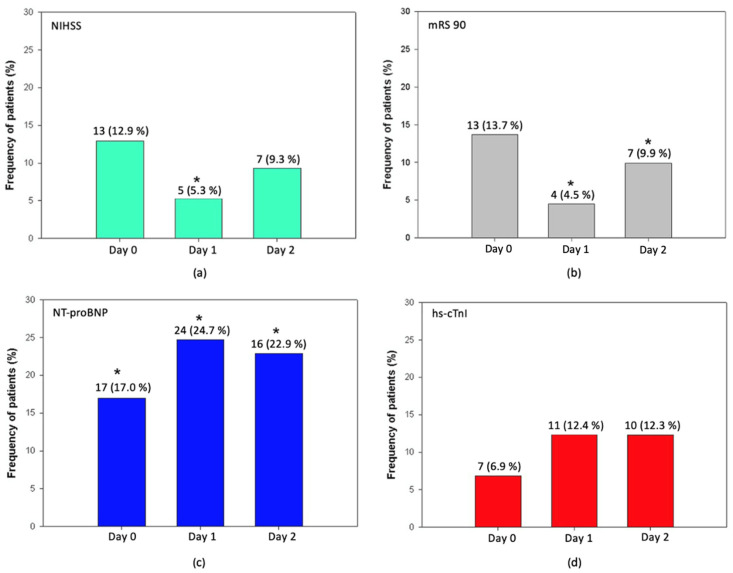
Relationship between lower TRAIL level (< 64 pg/mL) with markers of myocardial injury, stroke severity, and functional outcome in AIS. (**a**) relationship between lower TRAIL and moderate to severe stroke (NIHSS 16–42); (**b**) relationship between lower TRAIL and worse functional outcome or death (mRS at 90 day 5–6). (**c**) relationship between lower TRAIL and NT-proBNP elevation >125 pg/mL; (**d**) relationship between lower TRAIL and hs-cTnI elevation >53 pg/mL; The categorical values are given as frequencies and respective percentages. * Represents significant result.

**Figure 4 jcm-11-02552-f004:**
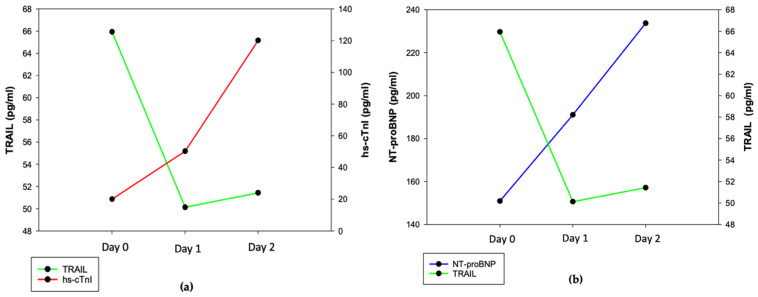
Relationship between TRAIL and markers of myocardial injury in patients after ICH. (**a**) Association between NT-proBNP and TRAIL; (**b**) association between TRAIL and hs-cTnI; mean values and SD: NT-proBNP day 0: 150.8 ± 143.4, day 1: 191.1 ± 158.9, day 2: 233.7 ± 276.8, TRAIL day 0: 65.9 ± 31.7, day 1: 50.1 ± 27.3, day 2: 51.4 ± 32.9, hs-cTnI day 0: 20.1 ± 23.9, day 1: 50.2 ± 87.8, day 2: 120.1 ± 349.1.

**Table 1 jcm-11-02552-t001:** Baseline characteristics of 120 included patients with acute cerebrovascular disease.

	Overall, n = 120	AIS, *n* = 104	ICH, *n* = 16	*p* Value
Baseline characteristics				
Age, Mean year (SD)	70.9 (12)	70.8 (11.8)	71.8 (13.5)	0.84
Male, n (%)	63 (52.5)	53 (51)	10 (62)	0.43
Arterial hypertension, n (%)	93 (77.5)	78 (75)	15 (93.7)	0.12
Smoking, n (%)	51 (42.5)	43 (41.3)	8 (50)	0.81
Dyslipidemia, n (%)	60 (50)	53 (51)	7 (44)	0.79
Diabetes mellitus, n (%)	29 (24.1)	24 (23.1)	5 (31.3)	0.53
Ischemic heart disease, n (%)	8 (6.7)	6 (5.8)	2 (12.5)	0.29
History of stroke/TIA, n (%)	13 (10.8)	10 (9.6)	3 (18.8)	0.38
Atrial fibrillation, n (%)	26 (21.7)	23 (22.1)	3 (18.8)	0.21
Renal insufficiency, n (%)	9 (7.5)	6 (5.8)	3 (18.8)	0.09
History of myocardial infarction, n (%)	4 (3.3)	3 (2.9)	1 (6.3)	0.44
Assessments				
Symptom duration				
<4.5 h, n (%)	93 (77.5)	81 (77.9)	12 (75)	0.75
<12 h, n (%)	115 (95.8)	101 (97.1)	14 (87.5)	0.13
NIHSS				
0 (No stroke symptoms)		0		
1–4 (Minor)		30 (28.8)		
5–15 (Moderate)		52 (50)		
16–20 (Moderate to severe)		16 (15.4)		
21–42 (Severe)		6 (5.8)		
mRS 90 days				0.02
0 (No symptoms) (%)		34 (34.7)	1 (6.3)	
1 (No disability despite symptoms) (%)		22 (22.5)	1 (6.3)	
2 (Slight disability) (%)		12 (12.2)	1 (6.3)	
3 (Moderate disability) (%)		6 (6.1)	1 (6.3)	
4 (Moderate severe disability) (%)		6 (6.1)	1 (6.3)	
5 (Severe disability) (%)		5 (5.1)	1 (6.3)	
6 (Dead) (%)		13 (13.3)	8 (57.1)	

AIS—acute ischemic stroke, ICH—intracerebral hemorrhage, TIA—transient ischemic attack, NIHSS—National Institutes of Health Stroke Scale, mRS—modified Rankin Scale.

**Table 2 jcm-11-02552-t002:** Relationship between lower TRAIL levels and ECG changes in AIS.

	N (%)	*p* Value
Arrythmias		
Atrial fibrillation	20 (21.3%)	0.13
AV block I.degree	8 (8.5%)	0.78
PVC	7 (7.4%)	0.04
LAH	4 (4.3%)	0.49
RBBB	4 (4.3%)	0.37
Sinus tachycardia	3 (3.2%)	0.75
Morphological changes		
QTc prolongation	16 (17.0%)	0.052
ST segment depression	11 (11.7%)	0.29
T wave inversion	10 (10.6%)	0.14
Flat T wave	6 (6.4%)	0.92
U wave	2 (2.1%)	0.89

RBBB—right bundle branch block, PVC—premature ventricular contraction, LAH—left anterior hemiblock. The categorical values are given as frequencies and respective percentages.

## Data Availability

Not applicable.

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
