# Peer review of "Tumor Necrosis Factor-Related Apoptosis-Inducing Ligand (TRAIL) in Patients after Acute Stroke: Relation to Stroke Severity, Myocardial Injury, and Impact on Prognosis"

_jcm, 2022, doi:10.3390/jcm11092552_

Round 1

Reviewer 1 Report

The study presented by Mihalovič and colleagues is original research paper. Authors present descriptive study on the level of TRIAL in blood samples taken from patients suffered either from ischemic or hemorragic stroke.

Main:

The novelty of the study is not clear. There are reports on TRIAL level in patients after stroke. Author cite them well. What is new in your study? Time-dependent changes?

Minor:

how TRAIL values change in healthy age matched control group of patients? Author are advised either to present original data or to cite published.

Fig 1 b is the repetition of the data already presented in Fig 1 a. There is no references to Fig 1b in the text, therefore there is no need to show it as a separate subfigure.

Line 198: It would be good if you can add couple of words describing NT-proBNP (what this peptided does? Why it is important to monitor its level in the blood?). People that are not specialist would benefit from this information

The same is for troponin I (line 201). Please add short description of its function in stroke pathophysiology.

Figure 3 and 4: no SD are provided for plots.

Author Response

Dear Reviewer,

thank you very much for your comments. 

For the main point:

  •  Decribe novelty of the paper. In our study we focused on acute phase obtaining results at admission, and on the first and second day to be able to see the dynamic changes in acute phase and see connection with stroke severity, short term outcome and also cardiac markers as studies suggest its strong connection in cardiovascular pathophysiology. Previously conducted studies for example Kang et. al. and Pan et al evaluated TRAIL single measurement which was obtained within 7 days of hospitalization after stroke onset. Moreover, there is limited amount of publications describing TRAIL in cardiovascular research especially stroke.

For Minor points:

  • matched healthy control group cited in Tufekci et al. follow up analysis
  • we decided to delete Figure 1b
  • as suggested we presented short description of both NT-proBNP and troponin
  • We provided mean values and SD for plots under the figure

Thank you very much for all your valid comments, i hope we adressed the points properly.

Sincerely,

Mihalovič M.

Reviewer 2 Report

I appreciate the opportunity to review the manuscript." Tumor necrosis factor-related apoptosis-inducing ligand (TRAIL) in patients after acute stroke: relation to stroke severity, myocardial injury and impact on prognosis".

In this single-centre prospective observational study in patients with acute ischemic stroke (AIS)(n=104) and intracerebral hemorrhage (n = 16), the authors evaluated the association between tumor necrosis factor-related apoptosis-inducing ligand (TRAIL) and biomarkers (NT-proBNP, hs-TnI), stroke severity (NIHSS), and neurological 90-day outcome (mRS).

The authors conclude that TRAIL is a predictor of stroke severity and poor outcome in patients after acute stroke. TRAIL was also negatively associated (correlated?) with NT-proBNP and hs-TnI at admission, after 24 and 48 hours of hospitalization. Furthermore, lower TRAIL levels (arbitrarily defined as <64 pg/ml) were associated with premature ventricular extrasystoles and borderline with the QTc interval.

The results, although interesting, should be interpreted with caution, as several issues need further clarification 

  • The conclusions partially correlate with the results found, as the statistical analysis may not be appropriate for the Statistical analysis is not sufficiently described. Not all of the results presented match the methods described.
  1. Did the author check the normality of the data? Data are presented as mean and standard deviation (line 120); however, differences between groups were tested using the Mann Whitney U test commonly used for continuous non-normally distributed or cardinal variables. Figure 1C shows a box-and-whisker plot with median, IQR outliers, and extreme
  2. What type of correlation analysis was used? The correlation coefficient (R) is not reported in the results and is not presented in Figures 1a and 1b. When reporting the results of this analysis, I suggest using a positive/negative correlation instead of a significant positive/negative association. Try to avoid 'significant'.
  3. When evaluating the association of TRAIL with NIHSS and mRS in AIS, were these variables adjusted?
  4. The cutoff value for lower plasma TRAIL levels was arbitrary on the basis of previous publications. No analysis was performed to determine the optimal cut-off point of TRAIL for neurological Theauthors can provide more data.

  • Study design and patients.
  1. It is not reported whether patients with AIS received reperfusion therapy (intravenous thrombolysis or mechanical thrombectomy).
  2. Exclusion criteria/results: No acute infection or severe hyperglycemia that may influence TRAIL was This should be added to the limitations.
  3. Ventricular tachycardia in the ECG was defined by the authors as longer than 5 s. It is unusual as it is defined as tachyarrhythmia with more than three beats of ventricular origin.
  4. The prolongation of QTc prolongation was defined as >0.47s for men and >0.48s for women. What recommendations were used as a source of this threshold?
  5. Table 1; How renal insufficiency was defined?
  6. Line 115. What parameters were measured during an echocardiographic examination and evaluated in the context of TRAIL changes? Line 137 only "new left ventricular regional wall motion abnormality" was reported

Results

Was TRAIL associated with mortality? Please comment.

Did TRAIL influence mRS on day 90 and NIHSS in the ICH group?

  1. Figure 3 b, c, d – no p-values are reported. If the result of the comparison is statistically nonsignificant (Line 207), it should be presented.
  2. Line 207 The authors wrote: "We observed an analogous pattern in patients after ICH with a negative association between TRAIL and NT-proBNP and hs-cTnI at admission and on day 1." No p-values are reported in the figure and text. Despite the graphical pattern in Figures 3 b, c, and d, the difference should be proved with statistical tests. If not, the study shows only the association between TRAIL and NT-proBNP in the AIS group.
  3. Table 1 intergroup differences between the AIS and ICH groups should be evaluated.
  4. Add numbers and percentages, or mean+/- SD, median (IQR), not only p values in sections2 and 3.3. In Section 3.1, do not repeat the data from Table 1 in section (percentages).
  5. Line 144 - "TRAIL level decreased in the first 24 hours of hospitalization and subsequently increased on day 2." "How this difference was evaluated, Figure 1 shows a difference between the AIS and ICH groups. Did the authors perform a parametric or nonparametric test for repeated measurements with post hoc analysis at three time points?
  6. Line 149 "Low TRAIL levels were significantly associated with the presence of DM 2. type (p=0.046) and smoking (p=0.026)" Smoking is not reported in Table 1.
  7. Line 175 "We didn't find a significant association between lower TRAIL levels and admission to the hospital later than 4,5 hours, 12 hours, or with wake-up stroke." Data not
  8. Tablle 2 is unclear. What does theTRAIL association p-value show? If there is an association with categorically defined TRAIL <64 pg/ml, I suggest showing differences between groups, not a pure prevalence of ECG changes in the whole group.
  9. The prevalence of PVC prevalence was very low in the whole group. Please comment.
  10. Figure 1B. Association between TRAIL levels and mRS at 90 days. What is the time point 0, 24 or 48 hours?

Introduction and Discussion

The bibliography, based on its Introduction and Discussion Section, is not up to date.  Most references are relevant to the study; however, they do not refer to studies younger than 5 years (ref. 21). In addition, 17 of the 26 references have more than ten years. In the Introduction, the involvement of TRAIL in the pathophysiology of cardiovascular brain disease is based on the study by Martin-Villalba published in 1999.

For example, TRAIL may exert its biological effect through death receptors (R1 R2), as stated in the introduction. Furthermore, it is worth discussing TRAIL-R3 and R4, which are decoy receptors. Kakareko K, et al. T. TRAIL and Cardiovascular Disease: A Risk Factor or Risk Marker: A systematic review. J Clin Med. 2021;10(6):1252.. doi:10.3390/jcm10061252

Author Response

Dear Reviewer,

First of all, thank you for your valuable comments. We are confident that Your suggestions increased quality of the paper.

As for results:

  1. Normality of data was tested (added to methods), Fig1 B is the output of kruskal-wallis with median output, IQR, however we decided to delete figure 1b as it duplicited in part with fig 1A.
  2. For Fig 1 for differences between groups we used man whitney test. 
  3. Variables were not adjusted. Descriptive statistics have mostly been used, and due to the small number of patients and published studies, we thought multiparametric methods can be misleading.
  4. Cut off value was decided from median value of our study group in accordance primarly with studies from Kang et al, Volpato et al. 

Study design and patients:

  1. We added reperfusion therapy to patients description.
  2. We added severe hyperglycemia and acute infection to limitations.
  3. We changed the criteria for more than 3 beats as. However, in our group there was no report on NSVT or ST.
  4. Before we used adjusted criteria for QTc prolongation from guidelines to avoid overdiagnosing, however we decided to determine and change it to prolonged QTc for men >450ms and for women >460ms. It didnt change the association.
  5. We used KDIGO criteria and patients with already diagnosed renal insuficiency in anamnesis.
  6. Echocardiography: Standart echocardiographic examination was performed, evaluation of LV motion using 17 segment system. However, because of occurrence only 2 new LV dysfunction, we did not correlated results to TRAIL changes. It is possible to perform evaluation.

Results:

  1. Yes, lower TRAIL <64pg/ml on first day of hospitalization was associated with mortality at 90 day. Added.
  2. In ICH group we did not observe association between lower TRAIL level and worse functional outcome on day 0 (p=0,58), day 1 (p=,24), day 2 (p=0,59), however the patients frequency is very low.

  3. As of Figure 3 and 4 we replaced figure 3 and added graphs for better clarity of information
  4. In Table 1 we added intergroup differences
  5. We decided to replace figure 1a and b.
  6. For repeated measurements we used nonparametric test, man whitney
  7. Smoking added to table 1
  8. Data for time of admission added.
  9. We edited table 2 for more clarity. The table shows ECG changes with categorically defined TRAIL <64 pg/ml in AIS group. We didnt ad ICH due to group frequency however it is possible to compare groups. The categorical values are given as frequencies and respective percentages.

  10. Prevalence of PVC. Evaluation of ECG changes was performed from obtained 12 lead ECG and monitoration. However, continual monitoration wasn’t performed in all patients for first 48 hours. Some patients with low NIHSS and good clinical condition were admitted to standart unit where only 12 lead ECG were performed what in results could end up in missing some patients with PVC.

Introduction and discussion

We revised the text and edited references. Unfortunately, number of studies published concerning TRAIL and cardiovascular research and especially stroke is very limited. 

Thank you very much, I hope we adresses and clarified your points.

Sincerely,

Mihalovic M

Reviewer 3 Report

The article may be interesting, although I think that some nuance can be made.
First of all, to point out in the introduction, the factor, onset symptoms of stroke, since it later appears in methods, and it is a very important factor, I recommend this citation: https://pubmed.ncbi.nlm.nih.gov/31909910 /

likewise, this sentence should be corrected for unspecific, line 132:
Almost all the patients were admitted within 12 hours of symptoms onset.

On the other hand, the sample of people with hemorrhage is very low, so the results obtained have to be softened, that is, the comparison with stroke has to be nuanced.
in table 1, indicate the value of p

It also indicates longitudinal measurements, which are not taken into account later, I suggest using ANCOVA to obtain the differences between groups, spss 28 provides non-parametric ANCOVA.

The results should be raised more structured to facilitate reading, and in turn, see if all the graphics are necessary.

Author Response

Dear Reviewer,

Thank you very much for valid comments, which helped to improve our article.

  • Citation added as reccomended to the introduction.
  • In our group except one patient all were admitted withing 12 hours, edited the information.
  • p value in table 1 added
  • We have made some editing regarding data presentation, graphs and data evaluation. Because of lower patients frequency for multiparametric method we used man whitney for intergroup difference.

Thank you very much.

Sincerely,

Mihalovic M

Round 2

Reviewer 2 Report

I thank the authors for their answers.

I have only some minor comments regarding the paper.

The heterogeneity of the studied population regarding the type of reperfusion therapy should be added to the limitations. The number of patients undergoing reperfusion therapy should be added to the abstract.

The author’s reply on prevalence of PVC should be added to the discussion or limitation section.

Lines 130-132: Please clarify to avoid repetitions “Parameters that did not follow a normal distribution were analyzed with the MannWhitney U test or the Kruskal-Wallis test and expressed as the median. Testing of differences between groups was performed by Student's t-test or Mann Whitney U test.”

Figure 4 seems to be doubled. Figure 1 covers an unnamed figure (1B?).

An abbreviation for acute myocardial infarction is rather AMI than AIM.

Line 196 missing ‘0” [(p=,24)]

Standardize the type and placement of periods in the bracket references and when reporting p-values.

Line 328 Please remove “independent” as only univariate analyzes were performed

Try to avoid ’significant’.

Author Response

Dear Reviewer,

thank you very much for additional points. We made adjustments according to the proposals.

Sincerely,

Mihalovic M

Reviewer 3 Report

The authors have made the requested changes.

Author Response

Dear Reviewer,

thank you very much for manuscript reviewing.

Sincerely,

Mihalovic M